# Formation and Characterization of Oregano Essential Oil Nanocapsules Applied onto Polyester Textile

**DOI:** 10.3390/polym14235188

**Published:** 2022-11-29

**Authors:** Carla Salinas, Manuel J. Lis, Luisa Coderch, Meritxell Martí

**Affiliations:** 1Institute for Advanced Chemistry of Catalonia (IQAC-CSIC), Jordi Girona 18-26, 08034 Barcelona, Spain; 2INTEXTER-UPC, Colom 15, 08222 Terrassa, Spain

**Keywords:** oregano essential oil, polyester, polymeric nanocapsules, nanoprecipitation, glycerol

## Abstract

Oregano essential oil was encapsulated in poly-ϵ-caprolactone nanoparticles by a nanoprecipitation method using glycerin as a moisturizer. Nanocapsule characterization was performed by measuring the particle size, colloidal stability and encapsulation efficiency using dynamic light scattering, UV–Vis spectrophotometry and scanning electron microscopy (SEM). The nanoparticles had a mean particle size of 235 nm with a monomodal distribution. In addition, a low polydispersity index was obtained, as well as a negative zeta potential of −36.3 mV and an encapsulation efficiency of 75.54%. Nanocapsules were applied to polyester textiles through bath exhaustion and foulard processing. Citric acid and a resin were applied as crosslinking agents to improve the nanocapsules’ adhesion to the fabric. The adsorption, desorption, moisture content and essential oil extraction were evaluated to determine the affinity between the nanocapsules and the polyester. The adsorption was higher when the citric acid and the resin were applied. When standard oregano nanocapsules were used, almost all of the impregnated nanoparticles were removed when washed with water. The moisture content was evaluated for treated and non-treated textiles. There was a significant increase in the moisture content of the treated polyester compared to the non-treated polyester, which indicates that the polyester hydrophilicity increased with an important absorption of the essential oil nanocapsules; this can improve fabric comfort and probably promote antibacterial properties.

## 1. Introduction

For decades, essential oils (EOs) have been widely used in medicinal therapies, pharmaceutical treatments, cosmetic applications and food conservation due to their antibacterial and antimicrobial properties [1]. EOs are natural organic compounds extracted from plants by different methods, such as distillation. Due to their hydrophobic nature and high volatility, they are often lipophilic [1,2]. Numerous studies have shown that EOs are highly efficient in preventing the growth of pathogens because of their excellent insecticidal and repellent activities [1,2,3,4,5]. Moreover, plant-based antimicrobial compounds possess several advantages over synthetic chemicals, as they are often non-toxic, have limited side effects and are environmentally friendly [6].

One of the most studied EOs for medical and pharmaceutical treatments is oregano essential oil (OEO) [7,8]. OEO, which is extracted from Origanum vulgare L., is well known for its antimicrobial and antioxidative activities and its high efficiency in treating infectious diseases [2,7]. These activities are due to carvacrol, thymol and two phenols, which are the main compounds in this substance [4,9]. Preus et al. (2005) studied the antimicrobial efficacy of several aromatic oils and reported that OEO was potent against two bacterial strains of Staphylococcus aureus, which are termed ATCC 14154 and ATCC 14775 [10]. The fungicidal properties of OEO have also been reported [11]. OEO is soluble in hydrophobic solvents, such as methanol and acetone. OEO possesses disinfectant and healing properties, which make it an excellent candidate for skin applications. Furthermore, pathogens cannot become resistant to this natural remedy, unlike conventional antibiotics.

As with many EOs, OEO evaporates and decomposes during drug formulation due to direct exposure to light, oxygen and heat, among other stimuli [3]. Polymeric encapsulation has been proposed as a method of preserving OEO due to its high volatility and rapid deterioration [2]. The development of multifunctional textile materials based on micro- and nanotechnologies represents one of the most competitive and dynamic areas of research worldwide, being a major factor in the sustainable development of the textile industry [12,13,14]. With essential oils being hydrophobic, formulations have been developed to protect their fragrance, bioavailability and pharmaceutical effects [15]. Drug encapsulation can offer good stability, enhanced bioavailability and controlled drug release [1].

Encapsulation of essential oils can be developed at the micro- or nanoscales. Microencapsulation [16] and nanoencapsulation of essential oils [17] provide several benefits, including controlled and targeted release of the encapsulated essential oils, decreasing the volatility of the oils by enhancing their shelf life, changing the aroma effect to customize the specific application of the product and increasing the stability of microencapsulated materials toward oxidation or deactivation due to environmental effects [18]. Nanoparticles (NP) are colloid-sized particles with diameters ranging from 10 to 1000 nm, whereas microcapsules have diameters between a few micrometers and a few millimeters [19,20]. Nanoscale materials are advantageous in microcapsules because they have a higher specific surface, which allows for a more efficient fragrance loading in the cavity of the carrier, while optimizing the interaction with the fragrance and its sustained release [21]. However, concerns regarding the accumulation of microplastics in the environment and pending regulations by the European Chemicals Agency (ECHA) have challenged researchers to design polymeric NPs that can demonstrate high adsorption to textiles and low or no desorption during scrubbing and recurring washing.

Recently, polymeric nanoparticles (NPs) have been developed to encapsulate EOs, providing them with good stability, controlled delivery, enhanced bioavailability and improved efficacy [1,7]. Nanoprecipitation is a simple and reproducible technique suitable for encapsulating hydrophobic materials. Two miscible phases are needed: an organic phase and an aqueous phase. The organic phase is composed of a polymer solution and an EO mixture dispersed in an organic solvent. Polymers such as polylactide (PLA) or poly-e-caprolactone (PCL) can be synthetic or natural [22,23]. Nanoemulsions are formed by a dispersed and a continuous phase. They do not form spontaneously when the two equilibrium phases are mixed, and their properties depend on the preparation method. In this study, nanocapsules were prepared in O/W solutions. The organic phase, which is composed of a polymer, OEO and a surfactant, is added to an aqueous solution to form the nanocapsules [24]. Surfactants are amphiphilic molecules that contain hydrophilic and hydrophobic groups; therefore, they can absorb interfaces and reduce surface tension. When choosing a surfactant for an emulsion, the HLB value should be considered. If the HLB value is high, it is suitable for O/W solutions. However, if the HLB value is low, it is suitable for W/O solutions.

Polymeric nanoparticles offer several advantages, such as controlled release of the drug and protection from degradation. However, they can suffer physical and chemical instabilities, which means that they can undergo aggregation, sedimentation or hydrolysis. In this study, PCL is proposed as a wall for drug encapsulation. PCL is a semicrystalline polymer with a glass transition temperature of approximately 60 °C and slow degradation. PCL is a preformed polymer suitable for double emulsion and nanoprecipitation processes. Indeed, PCL is insoluble in water and soluble in organic solvents, as well as it is nontoxic, biodegradable and biocompatible.

Moreover, OEO has been encapsulated to study its application to polyester textiles due to the numerous benefits that functionalized textiles can offer to the skin in contact with the textile. The antibacterial and antifungal properties of nanocapsules with OEO have already been mentioned, but emulsions can provide many more benefits, such as enhanced comfort. The thermophysiological comfort of a garment is determined by the fabric’s air permeability, moisture management and heat transfer properties [25,26]. Microencapsulated phase change materials have been widely used to enhance the thermal comfort of clothing [27]. Polyester (PES) is a highly hydrophobic textile with a moisture content of approximately 0.4% and a very low comfort level [28]. Therefore, hydrophilicity is a key parameter of the comfort of the fabric [26] and enhance emulsion impregnation onto textiles. PES fabric treated with glycerol was obtained, showing a significant improvement in moisture retention capacity and promoting comfort sensation [28]. In this research, glycerol-modified nanocapsules [22] were applied to PES fabrics, and then the moisture content was evaluated for treated and non-treated PES textiles. Additionally, the water sorption and desorption rates of different OEO nanocapsule (OEO-NC) formulations were compared.

The development of effective encapsulation is crucial to ensuring localized drug delivery and site-specific action. However, PES hinders emulsion impregnation because of its high hydrophobicity index and low affinity between the essential oils and the textiles. A resin and a crosslinking agent can be applied as fixing agents to improve the nanocapsule absorption onto PES textiles and their subsequent resistance to washing. The incorporation of a binder in this process can improve the fabric’s durability through washing and handling [29]. Crosslinking agents enable encapsulation properties and link different molecules together. Some examples compatible with PCL nanoparticles are tannic acid or citric acid, which can be applied before or after the emulsion phenomena [30]. The resin, however, can only be applied after the nanoparticle preparation, and a curing treatment is needed for the fixing agent.

The main goals of the current research were to prepare polymeric nanoparticles with glycerin based on oregano essential oil and apply them to a PES textile to promote EOE fixation and increase the hydrophilicity and comfortability of the fabric.

## 2. Materials and Methods

### 2.1. Materials

OEO, obtained from the Terpenic Lab, Barcelona, Spain, was the active material that was encapsulated. PCL (average Mn 45,000) was used as the wall polymeric material. Sorbitan monooleate (Span 80), which is a nonionic surfactant, was employed as the stabilizing agent. Glycerin (ACS reagent ≥ 99.5%) was used as a wetting agent, and all of these were obtained from Sigma Aldrich Chemicals Private Ltd. (Madrid, Spain). Center ES-FF resin, provided by Color Center (Barcelona, Spain), and citric acid, also obtained from Sigma Aldrich Chemicals Private Ltd., were the fixing agents, which had to be applied at 105 °C for 5 min after the nanocapsule impregnation. Acetone, purchased from MERCK (Merck, Darmstadt, Germany), was the organic phase solvent, and Milli-Q water was used for the aqueous phase solvent.

### 2.2. Methods

#### 2.2.1. Preparation of OEO-NCs by the Nanoprecipitation Method

OEO was encapsulated by the nanoprecipitation method, which is also called solvent displacement. It is usually employed to incorporate lipophilic drugs into carriers based on the interfacial deposition of the preformed polymer. Drugs, polymers and organic solvents are added to the aqueous solution, and nanoparticles are formed immediately by rapid solvent diffusion.

The nanoprecipitation process was reported by Fraj et al. 2019 [2]. First, the PCL, Span 80 (HLB = 4.3) and OEO were dissolved in an acetone solution while being magnetically stirred at 300 rpm at 50 °C until the PCL was completely melted. The polymer, surfactant and OEO ratios as a function of the acetone volume were 0.5% *w/v* polymer: acetone, 0.3% *w/v* surfactant: acetone and 1.20% *w/v* OEO: acetone, respectively. Then, the organic phase was added dropwise into an aqueous solution containing a mixture of Milli-Q water and a moisturizer (Tween 80 or glycerin) under magnetic stirring at 400 rpm. The moisturizer had a ratio of 0.15% *w/v* moisturizer: water. The relationship between the organic and aqueous phases was 30:70%. Then, 20 min after all of the organic phases were added, the acetone was finally removed using a rotary evaporator at 40 °C. The final volume was the same as the volume of the aqueous solution.

#### 2.2.2. Nanoparticle Characterization (Mean Size, Zeta Potential, Polydispersity Index and pH)

The mean diameter size, zeta potential (Z-Pot) and polydispersity index (PI) were measured by dynamic light scattering (DLS) using a Nano ZS Zetasizer ZEN3600 device (Malvern Instruments Ltd., Malvern, Worcestershire, UK). The DLS measures the Brownian motion of the particles and uses the information to determine the hydrodynamic size. The hydrodynamic size is defined as the size of a sphere that diffuses at the same rate as the particle that is being measured. Brownian motion is the random movement of the particles that results from collisions with solvent molecules; thus, smaller particles diffuse quickly. The rate of Brownian motion is quantified by the translational diffusion coefficient (D), based on the Stokes–Einstein equation. The Z-Pot is the difference in voltage at the double layer. If two particles are highly charged, then there will be no aggregation due to repulsive forces. To measure the Z-Pot, the samples must first be diluted (20 μL dispersed in 2 mL of deionized water). The pH was measured for each prepared emulsion with a pH meter (Radiometer Copenhagen).

#### 2.2.3. Encapsulation Efficiency (EE%)

The EE% of the OEO-NCs was determined by UV–Vis spectrophotometry (CARY-300, Agilent Technologies, California, EEUU). To achieve encapsulation efficiency, the nanoparticles must first be dissolved in an organic solvent, in which the nanocapsules precipitate [8]. OEO-NCs were dissolved in methanol solutions of various concentrations and vortexed. Then, the samples were filtered and centrifuged at 10,000 rpm for 30 min at 4 °C. Thereafter, the supernatant was extracted using a micropipette and analyzed for drug content using a UV–Vis spectrophotometer at a maximum absorption wavelength of 274 nm. Before the nanoparticle’s EE% was analyzed, a standard curve was created by diluting different essential oil concentrations in methanol. Finally, the sample drug concentrations were determined by using a calibration curve and applying the following indirect encapsulation efficacy formula (Equation (1)):(1)$$\%\;\mathrm{EE}=\frac{{\mathrm{OEO}}_{i}-{\mathrm{OEO}}_{s}}{{\mathrm{OEO}}_{i}}\cdot 100$$
where OEO_i_ is the initial amount of loaded drug and OEO_s_ is the amount of drug present in the supernatant analyzed by UV–Vis spectrophotometry.

#### 2.2.4. Application of Nanoparticles onto PES Fabric

The textile impregnation method was optimized by applying bath exhaustion and foulard processing techniques. The bath exhaustion was performed using a GLF1083 shaking water bath (Gesellschft fur Labortechnik mbit, Burgwedel, Germany) for 1 h at 40 °C with a 1/10 bath ratio (10 mL of NC solution for each gram of PES textile). The foulard method was carried out using a pad dry machine (Ernest Benz AG KLDHT). The PES fabrics were passed through the foulard and then squeezed by a pair of rollers at 1.5 Pa three times. An optimized impregnation was performed by combining the bath exhaustion followed by the foulard process with a pick-up of 95 ± 5%. To increase the OEO nanoparticle fixation, the two following strategies were used: addition of citric acid in the bath 10/1 (*w/w* NC/product) or addition of Center ES-FF resin in the bath followed by a curing treatment of 5 min at 105 °C. The textiles were dried in a set camera for 24 h at 20 °C and 65% humidity. Then, the fibers were weighed again to obtain the gained mass. Finally, the PES desorption was studied by rinsing the fibers three times with water (BR:1/50 at room temperature). Again, they were dried in the set camera for 24 h under the same conditions and weighed to obtain the loss mass.

#### 2.2.5. OEO Extraction from PES Fabric

The OEO extraction from the PES fabric was performed using methanol as the solvent, and the evaluation of the OEO content was performed by spectrophotometry using UV–Vis spectrophotometry. First, 2 × 2 cm pieces of the PES fabric were cut and placed in 20 mL vials of methanol. Then, the OEO extraction was performed under magnetic stirring at 400 rpm for 30 min and sonication for 15 min. Finally, the extracted OEO content was calculated based on the initial amount of fiber, the OEO-NCs absorbed onto the textile, and the concentration calculated by using the previously prepared calibration curve. This procedure was performed in triplicate.

#### 2.2.6. Moisture Content

The moisture content was studied on nanocapsule-impregnated textiles using the weight differences before and after drying the fibers in an oven. A sample of 0.5 g was maintained in a conditioned room (20 ± 2 °C and 65 ± 5% RH) for at least 24 h before being weighed and subsequently dried in an oven at 105 °C for 24 h. After the sample was cooled in a desiccator under a P_2_O_5_ atmosphere, it was weighed again, and the moisture content was calculated as a percentage in triplicate.

#### 2.2.7. SEM

The morphology of the nanocapsules on the emulsion and when applied to the PES fabric was observed by scanning electron microscopy (Hitachi instrument TM-4000Plus, Tokyo, Japan), which is a low-vacuum device combined with a high-sensitivity backscattered electron detector. This device enables nonconductive samples to be imaged without requiring a coating. The nanoparticles were dried onto aluminum mounting pins for 24 h before the analysis, and the PES fabric was evaluated directly.

## 3. Results and Discussion

### 3.1. Optimization and Characterization of OEO-NCs

The OEO nanocapsules were prepared using the nanoprecipitation method, which is very common for encapsulating hydrophobic drugs. This preparation method has been reported in a previous study [2] and is described in the experimental section. Tween 80 was previously used as a wetting agent and stabilizer [2]. In the current experiment, Tween 80 and glycerin were studied and compared as possible moisturizers for the OEO-NCs with the aim of reducing aggregation, creaming, coalescence, phase inversion and Ostwald ripening effects. In addition, the presence of glycerin is expected to promote hydrophilicity in the PES fabric when the OEO-NCs are applied.

Both glycerin and Tween 80 are nonionic substances that enhance the storage stability of emulsions through displacement of the surface. The OEO-NCs were prepared with these two different moisturizers by a nanoprecipitation process under constant conditions with the aim of obtaining particles with a small size distribution, high encapsulation efficiency and good emulsion stability. The OEO nanoparticles were characterized by DLS, where the mean particle size, PI and Z-Pot were determined. The encapsulation efficiency was determined by UV–Vis spectrophotometry to evaluate the OEO content at 274 nm.

Table 1 shows how the particle size and PI are similar for the glycerin and Tween 80 formulations. However, the Z-Pot value for the glycerin samples was substantially higher than that of the Tween 80 samples. Hence, it can be assumed that glycerin can provide better stability and improve the physical properties of OEO-NCs. It has been reported that the emulsion stability and wettability of the particles are strongly dependent on the pH [31] and that increasing the pH of the emulsion causes higher absolute values of the Z-Pot, resulting in higher stability.

Moreover, the encapsulation degree was evaluated for the two OEO-NCs. While the Tween 80 OEO-NCs presented an EE% of 51%, the glycerin OEO-NCs presented an EE% of 75.4% (Table 1). The encapsulation efficiency can be influenced by the partition coefficient of the target molecule in the solvents, the size distribution of the nanocapsules or the preparation method used. Even though the EE% was below 90%, 75% was good enough for the nanoprecipitation method.

Therefore, the nanocapsules with glycerin (OEO-NCGL) were prepared and characterized in triplicate for application to the PES fabrics. Colloidal dispersions were obtained with opalescent and turbid single-phase aspects. This result suggests that the nanoprecipitation method and the conditions are suitable for obtaining OEO-PCL-GL emulsions in equilibrium. The nanocapsules presented a mean size value of 235 ± 3 nm and a PI of 0.18 ± 0.02. As the obtained PI was lower than 0.2, the nanocapsules did not form aggregates and had a narrow mean size distribution, which means that the nanoparticles were homogeneous and exhibited monomodal behavior.

A qualitative study of the morphology and surface characteristics was performed using SEM. This technique enables the visualization of the sizes and shapes of the nanocapsules. In addition to showing the spherical shape of the OEO-NC-GL, it is possible to correlate the DLS mean size results to the SEM analysis. As shown in Figure 1, the nanoparticles were heterogeneously distributed and exhibited a certain degree of agglomeration. As the emulsion stability is dependent on time, the difference between the DLS and SEM results can be justified because the DLS analysis was performed just after the nanocapsule production, while the SEM was carried out after approximately fifteen days.

### 3.2. OEO-NC-GL Application to PES Textiles

#### 3.2.1. Optimization of the NC Impregnation Method to PES Textiles

Four different methods of applying nanocapsules to PES textiles were studied. Previous studies [2] optimized the bath exhaustion conditions at 1 h and 40 °C. In the foulard process, the conditions were determined by maximizing the absorption percentage without damaging the fabric since pressure exerts mechanical strength on the textiles. The bath exhaustion method yielded the lowest percent adsorption, providing only 0.20% pick-up. The foulard method showed a result of up to 0.44%, and the bath exhaustion combined with the foulard process yielded a 0.62% nanocapsule absorption. Considering that the bath ratio was 1/10 and the nanocapsules accounted for approximately 2% of the weight in the total volume, the low pick-up percentages were expected. To increase the amount of product applied, three foulard passes were performed after the bath exhaustion, obtaining the best result of approximately 1.3%.

As shown in Figure 2, the absorption percentage reached its maximum when the foulard and bath exhaustion methods were combined. Since PES is a hydrophobic fiber, it can be assumed that almost all of the adsorption occurred on the nanocapsules. In subsequent studies, the impregnation method will include both the bath exhaustion and the foulard process.

#### 3.2.2. Optimization of NC Fixation Method on PES Textile

An absorption/desorption study was performed with two crosslinking agents using the optimized impregnation method. As PES is a highly hydrophobic textile and the nanocapsules are suspended in an aqueous solution, a crosslinking agent or resin was applied to increase the adsorption efficiency. The goal was to improve the affinity between the textile and the nanocapsules by making the nanocapsules react with the OH-groups on the textile. Citric acid was chosen as the crosslinking agent because, when it dissolves in an aqueous solution, the carboxylic acid groups lose a proton, which increases the attraction between the emulsion and the textile. In contrast, the resin is composed of an acrylic styrene copolymer. This product is widely used for coating due to its outstanding pigment-binding properties. The film’s strength offers durability and resistance to removal by washing. Furthermore, the resin can give the fabric a finish of flexibility and hardness.

The results shown in Figure 3 reveal that when no crosslinking agent or resin is added, the fabric is capable of absorbing only 1.33 ± 0.23% of the emulsion. When the crosslinking agent was added, the absorption increased to 2.79 ± 0.34%, which is a notable improvement, but the best result was obtained with the addition of the resin, leading to a result of 3.43 ± 0.37%.

Another important factor to consider is the washing fastness. The goal was to achieve a textile capable of showing resistance to external agents, especially water, which is important for its permanent antimicrobial activity. Therefore, desorption was performed by rinsing the treated textiles three times with water.

After rinsing, a desorption of 1.03 ± 0.12% was observed for the OEO-NC-GL without crosslinking agents, 0.90 ± 0.16% was observed for the OEO-NC-GL mixed with the citric acid, and 1.22 ± 0.14% was observed for the OEO-NC-GL mixed with the resin. However, by comparing the adsorption/desorption ratios, it can be stated that the resin treatment is the one in which more of the OEO-NC-GL remains. Therefore, the rinsed textile emulsion with no fixing treatment retained only 0.33%, the citric acid emulsion retained 1.89%, and the resin emulsion retained 2.23%. Although the adsorption was significantly improved with the addition of a crosslinking agent and resin; these percentages are still low. Treatment of the textiles with ozone or plasma with oxidative gases should be pursued in further studies. The aim of these techniques is to improve the adsorption rate by modifying the fiber surface and the internal structure.

#### 3.2.3. OEO Extraction from PES Textiles

The main challenge when nanocapsules are applied to a textile is proving that the OEO-NC-GL has actually been impregnated. In the previous section, the adsorption rate was obtained from the weight difference; therefore, the final content of OEO present in the PES fibers was not revealed. In this section, the amounts of the corresponding OEO present in each treated fabric were obtained by extrapolating the gravimetry results from the previous section, taking into account the weight of the other actives in the nanocapsule formation. The OEO content present on the textile was also determined by extracting the OEO with an organic solvent, such as methanol, and the OEO content was determined by measuring the absorbance at 274 nm with a UV–Vis spectrophotometer (Table 2).

In both cases, the highest observed weight corresponded to the formulation containing the resin, and the lowest was the standard OEO-NC-GL. The results of the OEO extraction study matched the % *owf* values, which are shown in Figure 3, and the extrapolation of the OEO took into account the other activities of the formulation.

In all of the cases, the amount of OEO detected by gravimetry is slightly higher than that detected by the extraction, and this discrepancy is more pronounced for the samples treated with the citric acid and resin. This trend may be because the citric acid and resin significantly contribute to the weight in the adsorption study. In summary, when nanoparticles are mixed with a crosslinking agent, this compound could also be present in the fabric. Since textiles must be biocompatible, because most of them are used in dermal applications, it is important to keep in mind that the crosslinking agent cannot release toxic products, such as formaldehyde.

#### 3.2.4. Moisture Content

Moisture is an essential parameter for defining the properties of a fabric in terms of comfort [26]. In this study, the moisture content was evaluated for treated and non-treated PES textiles. Hydrophilicity is directly related to comfort, which is one of the important aspects of textile fabrics. Glycerin has wetting properties and can significantly increase the hydrophilic index [28,32].

Therefore, hydrophilic PES treated with OEO-NC-GL was compared to native PES fibers that had only been washed with water. The moisture content of the fibers treated with the resin, citric acid and standard OEO-NC formulations was also evaluated (Table 2).

The moisture content of the PES was evaluated gravimetrically at 65% RH and at room temperature. As shown in Table 2, the non-treated textiles had the lowest observed moisture percentages (0.641 ± 0.042%), and that of the PES fabric treated with OEO-NC-GL was higher than 2% in all cases. This significant difference is related to the fact that the emulsions were formulated with glycerin in the aqueous phase; thus, the glycerin enhanced the hydrophilicity of the PES and, therefore, increased the emulsion adsorption onto the textile. No differences were observed between the resin, citric acid and standard nanocapsules since all of the moisture percentages were similar. Therefore, the addition of a crosslinking agent was not correlated with the enhanced textile hydrophilicity.

#### 3.2.5. SEM in Treated Textiles

Scanning electron microscopy (SEM) was performed to visualize evidence of the addition of the nanocapsules to the PES textile. The SEM images are shown in Figure 4; there were no significant differences between the OEO-NCs of the textiles that had undergone different application methods. The SEM images of the three textiles showed nanocapsules attached to the PES fibers. Some of them had spherical shapes, and others had undetermined shapes, which were believed to be due to small agglomerations that may have occurred after drying.

## 4. Conclusions

The main goal of this research was to prepare polymeric nanoparticles with glycerin and oregano essential oil and apply them to a PES textile to promote OEO deposition and fixation and increase the hydrophilicity of the fabric. By qualitatively and quantitatively characterizing the nanoparticles, it can be concluded that the size and shape are appropriate to offer nanometric advantages in drug delivery with a high encapsulation efficiency, similar to previous reported results [2]. The main size of the particles was 235 ± 3 nm, the negative Z-Pot had a value of −36.3 ± 0.6 mV, and the EE (%) was 75.54 ± 3.78%.

The affinity between the textile, nanocapsules and aqueous medium was evaluated to achieve an elevated impregnation rate. As PES is a highly hydrophobic textile, nanocapsule retention is a challenge. Citric acid and a resin were mixed with the OEO-NC-GL to increase the affinity between the nanocapsules and the PES fibers. The best adsorption results were obtained when the resin was added to the emulsion (3.43 ± 0.37%). Moreover, when citric acid was added, a significant increase in the adsorption was observed (2.79 ± 0.34%). After the fabric was washed, a considerable number of OEO-NCs were released from the textile. However, 2.21% *owf* and 1.89% *owf* still remained on the PES fabrics. Other strategies would have to be explored to enhance the resistance to washing.

Glycerin was used as a wetting agent to dissolve in the aqueous phase to form the nanocapsules and increase the PES hydrophilicity. Consequently, the comfort of the fabric was enhanced as well. An important difference in the moisture content was observed between the treated and non-treated PES textiles. The moisture content of the PES was approximately 0.6%; however, it increased to 2.5% when the textile was impregnated with OEO-NC-GL with or without the fixing agent.

In conclusion, an upgraded methodology for forming nanocapsules with glycerin to encapsulate oregano essential oil has been optimized and characterized, resulting in a homogeneous formulation with monomodal behavior and high encapsulation efficiency. Application of the nanocapsules to PES was performed in the presence of citric acid or a resin, which enhanced their affinity to the fabric. In addition, the presence of glycerin in the nanocapsules promoted an increase in the hydrophilicity and, consequently, improved the fiber comfort. Nanoencapsulation research still has a long way to go, especially in the area of essential oil encapsulation. The nanoprecipitation process is difficult to scale, and the throughput is usually low. However, despite all of these drawbacks, drug encapsulation is one of the most promising strategies because it offers controlled drug release within a certain timeframe, side effects can be avoided, drug targets can be optimized and nanocapsules can enhance the bioavailability and stability of drugs.

## Figures and Tables

**Figure 1 polymers-14-05188-f001:**
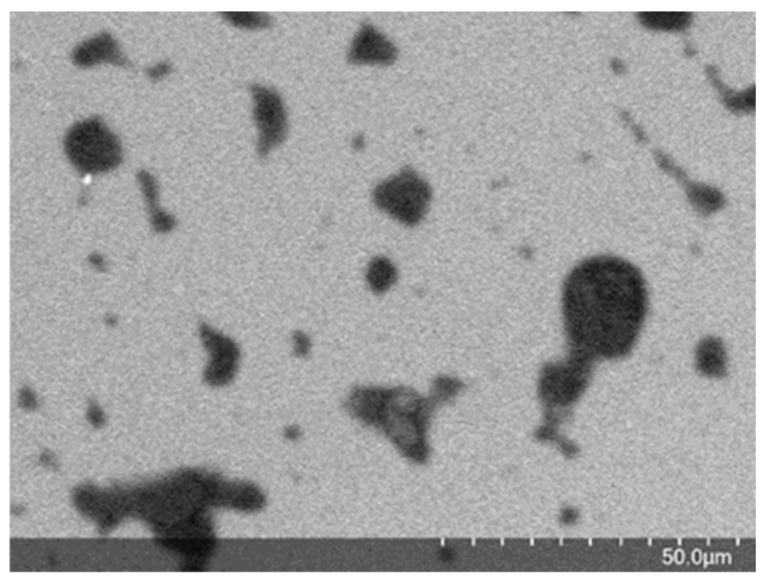
SEM image of OEO-NC-GL at 50 μm.

**Figure 2 polymers-14-05188-f002:**
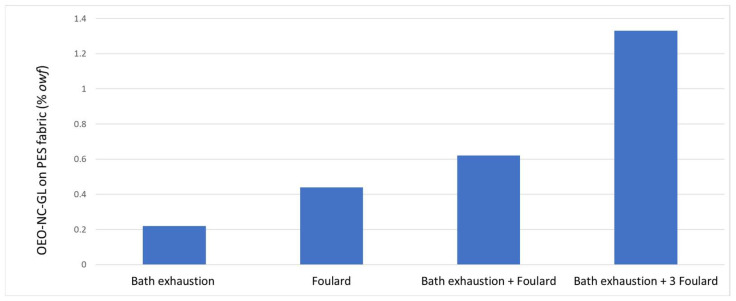
OEO-NC-GL adsorption onto PES textiles: comparison of three optimized impregnation methods. (*owf*: over weight of fiber).

**Figure 3 polymers-14-05188-f003:**
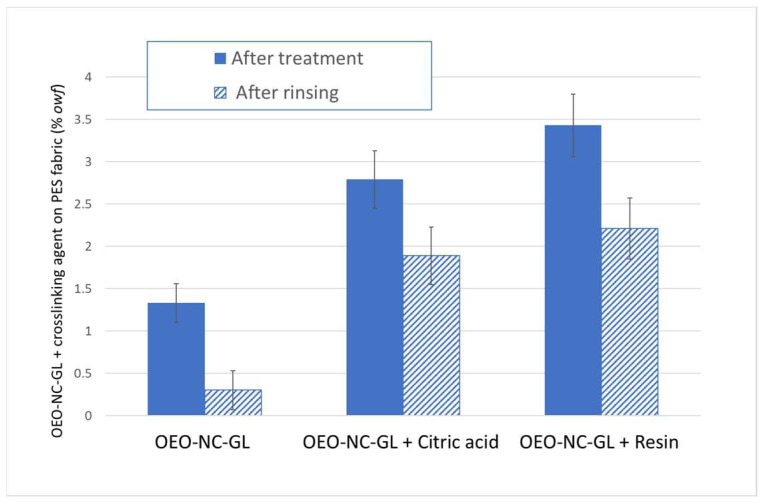
Adsorption and desorption results, % of product on PES fabric after OEO-NC-GL treatment and after rinsing. (*owf*: over weight of fiber).

**Figure 4 polymers-14-05188-f004:**
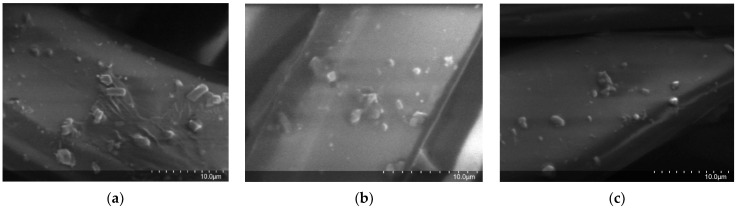
SEM images of OEO-NCs on PES textiles: (**a**) OEO-NC-treated PES, (**b**) OEO-NC + citric acid-treated PES (**c**) OEO-NC + resin-treated PES.

**Table 1 polymers-14-05188-t001:** Characterization of NCs prepared with Tween 80 vs. glycerin as moisturizers.

	Tween 80 NCs	Glycerin NCs
Mean Size (nm)	222 ± 2	235 ± 3
PI	0.21 ± 0.01	0.18 ± 0.02
Z-Potential (mV)	−24.6 ± 0.5	−36.3 ± 0.6
pH	5.5 ± 0.3	6.7 ± 0.7
EE (%)	51.3 ± 1.8	75.5 ± 3.8

**Table 2 polymers-14-05188-t002:** OEO content obtained from the gravimetrical and extraction methods and moisture content of the PES fabrics.

	OEO Content Gravimetry (% *owf*)	OEO Content Extraction (% *owf*)	Difference between Gravi. and Extraction (%)	Moisture at 65% RH
PES	--	--	--	0.641 ± 0.042
OEO-NC-GL PES	0.00127 ± 0.00015	0.00121 ± 0.00014	4.4%	2.453 ± 0.425
OEO-NC-GL + Citric ac. PES	0.00274 ± 0.00043	0.00256 ± 0.00093	7.0%	2.582 ± 0.569
OEO-NC-GL + Resin PES	0.00361 ± 0.00012	0.00330 ± 0.00054	9.5%	2.518 ± 0.489

## Data Availability

Not applicable.

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
