# Peer review of "Formation and Characterization of Oregano Essential Oil Nanocapsules Applied onto Polyester Textile"

_polymers, 2022, doi:10.3390/polym14235188_

Round 1
Reviewer 1 Report
1- The introduction needs improvement by adding more recently published references about the applications of encapsulation in the textile finishing as well as the encapsulation of essential oils and their application on textiles. The following are recommended:
a. Chirila, L., Constantinescu, G. C., Danila, A., Popescu, A., Constantinescu, R. R., & Săndulache, I. M. (2020). Functionalization of textile materials with bioactive polymeric systems based on propolis and cinnamon essential oil. Industria Textila, 71(2), 186-192.
b. Raza, S. M. O., Zehra, B., Usama, N. M., Kashif, I., Muhammad, Z., & Danmei, S. (2022). Modelling method to evaluate the thermo-regulating behaviour of micro-encapsulated PCMs coated fabric. Industria Textila, 73(1), 3-11.
c. https://doi.org/10.1108/RJTA-04-2022-0046
2- Figures 2 and 3, the title of the vertical axis needs revision.
3- The fastness against washing and rubbing is missing in this study. It is recommended to evaluate and compare the fastness properties as well.
4- FTIR spectroscopy may be helpful to confirm the presence of the EO on the prepared fabric. the SEM images cannot prove the existence the EO on the fabric alone.
5- As it is stated in the title, the aim has been to prepare a bifunctional polyester fabric. however, no test has been done to confirm its functionality. Antibacterial test may help in this case.
Author Response
Please find enclosed the answer to reviewer 1 as attachment.

Reviewer 2 Report
Dear Authors,
Please see my comments below:
(1) What is the novelty of this work?
(2) Introduction is too small and does not engage readers in why this work is so important and why it's needed. There are just 14 references. I am pretty much sure enough work has already been done in this field. Please include previous work and let readers know why your work is important and will contribute to this field.
(3) SEM shows only a few nanoparticles or agglomerate particles over the fabric. How do you ensure you get total coverage?
(4) What is the total weight % of nanoparticles stuck on the fabric?
(5) Have you compared the contact angle of fabric before and after nanoparticle adhesion?
(6) Is there any other technique such as TGA etc. you can use to show percentage of weight of polymeric nanoparticle attached on the fabric.
Author Response
Please find enclosed the answer to reviewer 2 as attachment.

Round 2
Reviewer 1 Report
Accept
Reviewer 2 Report
Dear Author,
The author has incorporated the said corrections. The manuscript is recommended for publication